**www.cambridge.org/ext**

Angiosperms; anthropogenic extinctions; extinction legacies; mass extinction; land plants

**Corresponding author:**
Peter Wilf;
Email: pwilf@psu.edu

# The end-Cretaceous plant extinction: Heterogeneity, ecosystem transformation, and insights for the future

Peter Wilf[1] 📷 , Mónica R. Carvalho[2] and Elena Stiles[3]

[1]Department of Geosciences and Earth and Environmental Systems Institute, Pennsylvania State University, University Park, PA, USA; [2]Museum of Paleontology and Department of Earth and Environmental Sciences, University of Michigan, Ann Arbor, MI, USA and [3]Department of Biology, University of Washington, Seattle, WA, USA

## Abstract

The Cretaceous–Paleogene (K–Pg) mass extinction was geologically instantaneous, causing the most drastic extinction rates in Earth's History. The rapid species losses and environmental destruction from the Chicxulub impact at 66.02 Ma made the K–Pg the most comparable past event to today's projected "sixth" mass extinction. The extinction famously eliminated major clades of animals and plankton. However, for land plants, losses primarily occurred among species observed in regional studies but left no global trace at the family or major-clade level, leading to questions about whether there was a significant K–Pg plant extinction. We review emerging paleobotanical data from the Americas and argue that the evidence strongly favors profound (generally >50%), geographically heterogeneous species losses and recovery consistent with mass extinction. The heterogeneity appears to reflect several factors, including distance from the impact site and marine and latitudinal buffering of the impact winter. The ensuing transformations have affected all land life, including true angiosperm dominance in the world's forests, the birth of the hyperdiverse Neotropical rainforest biome, and evolutionary radiations leading to many crown angiosperm clades. Although the worst outcomes are still preventable, the sixth mass extinction could mirror the K–Pg event by eliminating comparable numbers of plant species in a geologic instant, impoverishing and eventually transforming terrestrial ecosystems while having little effect on global plant-family diversity.

## Impact statement

The impact of an asteroid the size of San Francisco with Yucatán, Mexico at the end of the Cretaceous period, 66 million years ago, set a devastating series of events in motion. The massive species losses, popularly known as the "dinosaur extinction," eliminated an estimated 75% of land and sea species. The end-Cretaceous is highly relevant to modern-day projected extinctions as the only event in Earth's history that destroyed environments and wiped out life worldwide in a geologic instant. Land plants are the foundation of life on land, but suitable fossil-plant collections from the critical time interval are rare and geographically biased, leading to debate about whether there was any significant global plant extinction. We review new records of plant fossils from North and South America, which indicate significant (>50%) plant-species losses in each area, consistent with mass extinction. The different regions demonstrate heterogeneous responses, which are plausibly related to variations in distance to Mexico and the severity of the impact winter caused by atmospheric particles. Freezing the tropics, the most biodiverse region, may have generated devastating species losses. After the disaster came transformations of terrestrial ecosystems that define life on land today, including dramatic crown-group radiations, the rise to true dominance of flowering plants, and the birth of hyperdiverse tropical rainforests. Although it is not too late to avert the worst outcomes, projected losses of plant species in the near future are similar to estimates from the end-Cretaceous extinction. Ecological transformations will eventually follow, in all likelihood, too late to benefit humans.

## Introduction

Critical time intervals provide insights into the projected "sixth mass extinction" from anthropogenic disturbances (Wake and Vredenburg, 2008; Barnosky et al., 2011; Wing and Currano, 2013). However, the 66.02 Ma Cretaceous–Paleogene (K–Pg) mass extinction (here, KPgE) or "dinosaur extinction" stands above all known biotic crises for its suddenness. The KPgE destroyed approximately 75% of marine species and similar numbers of land animals in a geologic instant (see Jones et al., 2023 for a recent summary). Of the "Big Five" mass extinctions (Raup and Sepkoski, 1982; Marshall, 2023) and other critical intervals (Kiessling et al., 2023), only

the KPgE is directly comparable with the modern day for its speed of killing and environmental destruction (Barnosky et al., 2011).

Since Alvarez et al. (1980) proposed their impact-extinction hypothesis for the KPgE, the supporting evidence has grown overwhelmingly (Schulte et al., 2010; Hull et al., 2020). The unique cause of the KPgE was the Chicxulub bolide impact in Yucatán, Mexico at 66.02 Ma. The collision generated a global event horizon, an era boundary preserved at hundreds of sites from land to the deep ocean (Schulte et al., 2010; Morgan et al., 2022). Owing to recent drilling, the 200-km-wide Chicxulub crater's structure is known in outstanding detail, providing a highly resolved history of the grim "first day of the Cenozoic" (Gulick et al., 2019; Morgan et al., 2022). The crater's morphology is comparable to the largest observed in the solar system (Morgan et al., 2022). The estimated impact energy is $10^{23}$ joules, and returning tsunamis brought charcoal from distant forest fires back into the crater (Gulick et al., 2019). The sky went dark for months to years from dust, iron-oxide nanoparticles, aerosols, and $10^{14-15}$ g of black carbon from the target rock entering the upper atmosphere (Vajda et al., 2015; Lyons et al., 2020), thus freezing the surface, stopping photosynthesis, and causing the bulk of the mass extinctions.

Land plants are the foundational macroorganisms of terrestrial ecosystems, and their fate at the KPgE has drawn attention since well before 1980 (Dorf, 1940; see Nichols and Johnson, 2008). However, plants show very different extinction patterns from animals, partly because of their dramatically different life histories (Traverse, 1988; Wing, 2004; Cascales-Miñana and Cleal, 2014). The plant-fossil record since the Late Cretaceous is dominated by angiosperm leaves, which are well suited for species-level quantitative analyses in regional studies. However, most are not formally or accurately described and cannot be easily assigned to higher taxa, restricting spatial comparability (Wilf, 2008). The result is a sharp contrast between high-resolution regional stratigraphic studies, which usually show significant plant-species extinction and ecological shifts, and global syntheses or phylogenetic analyses, which show no losses of plant families or major clades. This situation has led to a debate regarding whether there was any significant K–Pg extinction for plants (Cascales-Miñana and Cleal, 2014; Nic Lughadha et al., 2020; Thompson and Ramírez-Barahona, 2023). Several previous reviews have covered the general topics of K–Pg plant extinctions, the vital contributions of palynology, and the rich history of investigations (Wing, 2004; McElwain and Punyasena, 2007; Nichols and Johnson, 2008; Vajda and Bercovici, 2014).

Here, we focus on the improving macrofossil (and associated palynological) records of the KPgE from the Americas and the increasing understanding of heterogeneity in the event's severity, ecosystem effects, and legacies. Despite advances in many regions, only the Western Hemisphere has extensive, stratigraphically well-constrained collections of latest Cretaceous and early Paleocene plant macrofossils (e.g., Nichols and Johnson, 2008; Vajda and Bercovici, 2014). We find significant evidence for species losses of land plants that fulfill any reasonable definition of mass extinction and provide instructive analogs for the near future.

## Heterogeneity

General regional differences in the K–Pg plant extinction and recovery are well known (Wolfe, 1987; Nichols and Johnson, 2008; Vajda and Bercovici, 2014). However, a pronounced North American sampling bias exists for nearly all significant collections of Cretaceous (K floras) and Paleocene floras from well-dated stratigraphic sections with well-defined K–Pg event layers (Nichols and Johnson, 2008). The Williston Basin in the northern Great Plains, USA, paleolatitude ca. 50°N, remains by far the best-studied area (Johnson et al., 1989; Johnson, 2002).

Several factors are likely to promote heterogeneity. There is a global pattern of increasing K–Pg event layer thickness and sediment disturbance with proximity to the Chicxulub crater (Schulte et al., 2010). Thus, the first effect usually considered is the distance from ground zero and the proximal effects of shockwaves, tsunamis, and large ejecta. A second factor highly relevant to plants is maritime and latitudinal buffering of the impact winter (Figure 1; Bardeen et al., 2017; Brugger et al., 2017; Morgan et al., 2022). Both distance and buffering gradients predict large extinctions in western North America and the Neotropics, especially if the tropics froze, and less severe extinctions in the maritime areas of temperate Gondwana, where there is a growing list of survivor taxa (e.g., McLoughlin et al., 2011; summarized in Wilf et al., 2013). Biotic

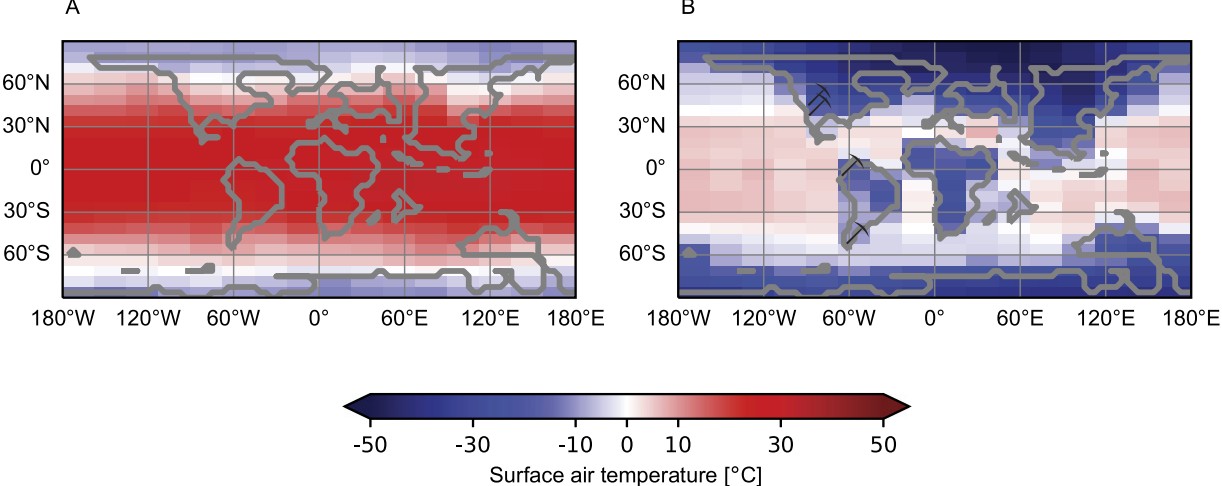

**Figure 1.** Modeled surface air temperatures (A) before and (B) during the coldest impact-winter year, 3 years after the Chicxulub impact; pickaxe icons indicate the principal fossil floras discussed in the text from (north to south) the northern Great Plains, southern Rockies, Colombia, and southern Argentina. Drafted by J. Brugger, using model output from Brugger et al. (2021), with permission.

variables include the diversity and traits (Berry, 2020; Butrim et al., 2022) of pre-impact and recovery floras.

## North America

In the Williston Basin near Marmarth, North Dakota, more than 160 plant-macrofossil localities spanning ca. 67.4–65.2 Ma were placed in precise stratigraphic positions measured to the well-preserved K–Pg event bed (Johnson et al., 1989; Johnson, 2002). The total number of paleobotanical specimens exceeds 22,000, with a significant representation of K and Paleocene floras. A conservative, robust estimate of macrofloral species extinction from these collections is 57%, based only on the species present in the upper-most 5 m (approximately 70 Kyr) of Cretaceous strata (Wilf and Johnson, 2004; Figure 2A). No other location has comparable sampling density and temporal precision, which must be considered when comparing extinction percentages (Figure 2B). Similar patterns of severe plant-species extinction, low early Paleocene diversity ("disaster floras"), and ca. 10 million years of recovery are known from significant but less complete datasets throughout the northern Great Plains (Wing et al., 1995; Hotton, 2002; Nichols and Johnson, 2008; Wilson et al., 2021). The regional Paleocene floras are well understood systematically and include many lineages that remain abundant in living north-temperate floras, such as taxodioid Cupressaceae, Cornales, Fagales, and Platanaceae (Brown, 1962; Hickey, 1977; Manchester, 2014).

The Southern Rockies present an emerging record of early Paleocene sites, several of which have much higher plant diversity than the northern Great Plains; however, few latest Cretaceous floras are known for comparison. The region is closer than the Williston Basin to Chicxulub, refuting distance from the crater as an explanation for the elevated Paleocene diversity and pointing to latitudinal effects combined with lower continentality (Figure 1B). Some sites in Colorado and New Mexico are said to preserve "tropical" forests (Johnson and Ellis, 2002; Flynn and Peppe, 2019), despite their location in temperate latitudes. This use of "tropical" to refer to warm past climates at middle latitudes is common in paleontology; however, the term is best applied only to the latitudinal tropics (<23.5°), where insolation and its effects on

plant life are far more significant (e.g., Jaramillo and Cárdenas, 2013).

Flynn and Peppe (2019) reported diverse early Paleocene (<350 Kyr after the K–Pg) floras from the San Juan Basin in New Mexico; there is no event horizon preserved, nor are K floras available for comparison. These assemblages represent frost-free conditions but lack characteristic tropical plant taxa (e.g., Carvalho et al., 2021). Instead, they appear to preserve a mixture of warm-temperate lineages similar to those found throughout the Paleocene north temperate zone, including species of Cupressaceae, Platanaceae, and Cornales. Further north, the Raton Basin of New Mexico and Colorado preserves a well-defined K–Pg event bed and was the primary location of early work on K–Pg plant extinctions following the publication of the Alvarez hypothesis (Tschudy et al., 1984; Wolfe and Upchurch, 1986). The area has received renewed attention, particularly concerning the basalmost Paleocene floras (Berry, 2019, 2023).

A rich macrofloral record comes from over 150 Late Cretaceous and Paleogene sites in the Denver Basin, Colorado (Johnson et al., 2003). The sites were stratigraphically positioned using a detailed basin age-depth model anchored to a well-defined K–Pg event horizon exposed at West Bijou Creek and found in the subsurface in the Kiowa Core (Barclay et al., 2003; Raynolds et al., 2007; Clyde et al., 2016). A showcase geochronology study combined paleomagnetic stratigraphy and U–Pb dating of a series of volcanic ashes at West Bijou Creek, constraining the age of the K–Pg boundary to 66.021 ± 0.024 Ma (Clyde et al., 2016).

We note that the exact age control of the KPgE represents decades of advances in geochronologic methods and inter-laboratory cooperation (Gradstein et al., 2012), in contrast to the continuing absence of geological benchmarks and standards that are needed to align molecular divergence dates with the inter-national geologic time scale (Wilf and Escapa, 2016). We remain skeptical of the substantial molecular-dating literature that assigns whole genome duplications (WGDs) in many plant lineages to an algorithmically-approximated "K"-"Pg" "boundary," implying that WGDs were beneficial for survival or recovery (e.g., Vanneste et al., 2014; Lohaus and Van de Peer, 2016; Koenen et al., 2021). This work also used few well-defined geological or phylogenetic criteria

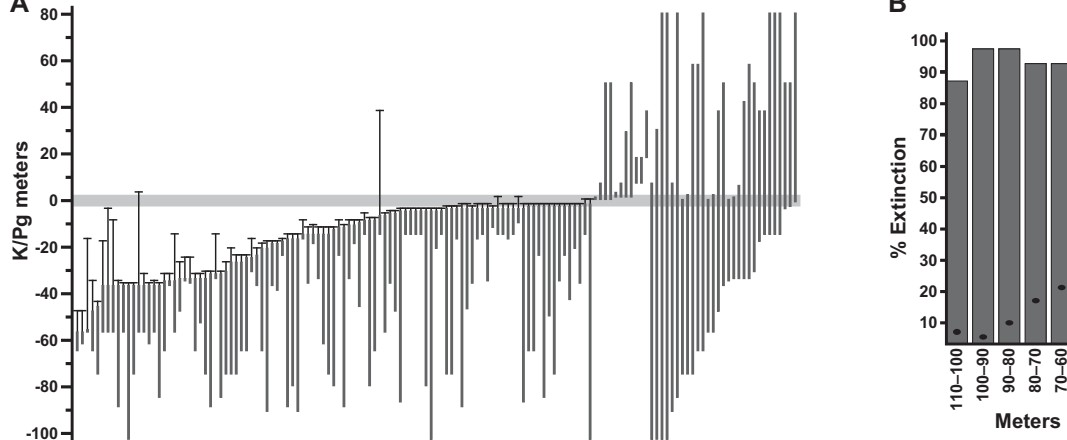

**Figure 2.** Plant extinction in the Williston Basin, North Dakota. (A) Ranges in the composite section of 141 macrofossil species occurring in more than one stratigraphic bin, with 99% confidence intervals on last appearances, relative to the K/Pg event horizon recognized from multiple indicators. The conservative estimate of 57% species extinction is based on this analysis, using only the last appearances in the uppermost 5 m of Cretaceous strata (Wilf and Johnson, 2004). The observed extinction increases if the window is shifted down, as demonstrated in (B), showing simulated extinctions and species richness from the same dataset (singletons included) when based on discrete 10-m bin windows (Stiles et al., 2020). Modified from (A) Wilf and Johnson (2004) and (B) Stiles et al. (2020).

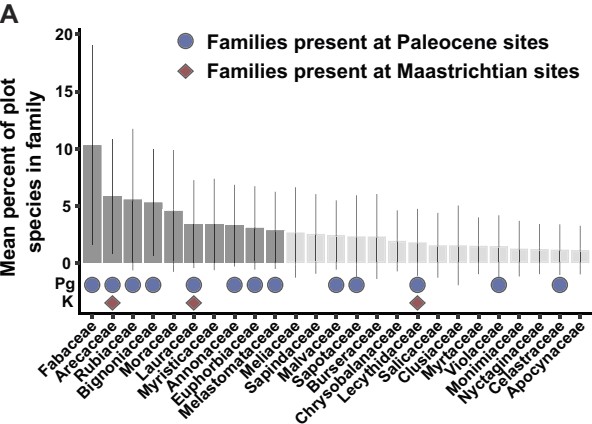

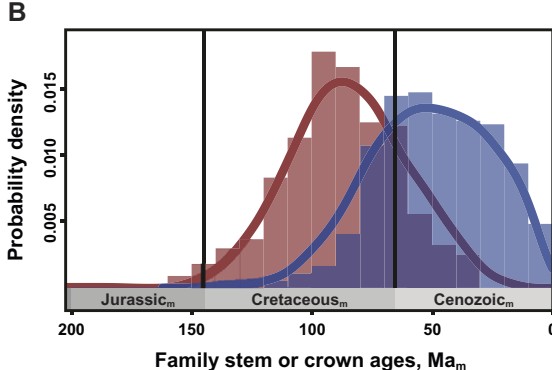

**Figure 3.** The post-KPgE rise of extant angiosperm families, seen in Colombian macrofossils and a molecular-phylogenetic reconstruction of stem- vs. crown-group age distributions. In (A), bars show proportions of species (with standard deviations) in a family in a compilation of 72 Neotropical vegetation plots; darker bars indicate the top-10 families that account for half of total stems. Red diamonds mark families present in a latest Cretaceous flora; blue circles show families recognized in two Paleocene floras. Modified from Carvalho et al. (2021). (B) Frequency distributions and 95% kernel density curves of reconstructed mean stem (red) and crown (blue) angiosperm family ages, from an analysis calibrated using 238 well-constrained fossils. Subscript "m" indicates age estimates derived from molecular analyses rather than direct geological dating. Modified from Ramírez-Barahona et al. (2020), with permission.

for fossil calibrations (contra Sauquet et al., 2012; Ramírez-Barahona et al., 2020) and followed the widespread practice of appropriating the language of geochronology despite the absence of analyzed rocks, thereby conflating different sources of evidence. At a minimum, molecular age estimates should use a distinctive labeling system (e.g., Figure 3B).

Returning to the Denver Basin, many of its early Paleocene floras are similar in composition and have low species diversity, like those of the northern Plains. However, several intriguing exceptions exist (Johnson et al., 2003), including the Corral Bluffs and Castle Rock floras. The Corral Bluffs (Lyson et al., 2019) exposures contain a well-dated sequence of ca. 66.2–65.2 Ma floras that co-occur with well-preserved mammalian remains, allowing direct correlations between floral recovery stages and increases in mammalian body size. The event horizon is not preserved, but the ages of the fossils are well constrained from a U–Pb dated ash layer, paleomagnetic stratigraphy, and a characteristic earliest Paleocene fern-spore spike. The sampling of K floras at Corral Bluffs is limited but sufficient to show a loss of more than half of the angiosperm species, consistent with the Williston Basin. Taxonomic work is at an early stage, but an exciting occurrence is the oldest definitive fossil legume fruit (Fabaceae); most elements, including Juglandaceae, Platanaceae, and palms, appear to be consistent with typical early Paleocene floras.

The ca. 63.8 Ma Castle Rock flora (Johnson and Ellis, 2002; Ellis et al., 2003; Erdei et al., 2019) was the first to break the mold of low-diversity, homogenous early Paleocene assemblages. The estimated richness of more than 100 species at Castle Rock is several times that of typical Paleocene floras from the western USA. Variable composition at the local scale (beta diversity) and prevalence of notably large leaves with drip tips indicate warm and humid conditions, leading to the flora's initial description as a "tropical rainforest in Colorado" (Johnson and Ellis, 2002). Large angiosperm leaves are easily fragmented and rarely preserved as fossils (e.g., Merkhofer et al., 2015), and they only occur in the modern world in humid, warm habitats (Wright et al., 2017). Thus, Castle Rock represents a minimally transported assemblage that records small-scale compositional variation and a warm, at least seasonally wet environment (Ellis et al., 2003). These observations led to the

hypothesis that vegetation adjacent to the Laramide Front Range received abundant orographic rainfall, which fostered high floral diversity (Johnson and Ellis, 2002). An alternative idea would be that Castle Rock more faithfully preserves large leaves and diversity than other early Paleocene sites but does not represent fundamentally different source vegetation similar to a tropical rainforest. Taxonomic studies of the Castle Rock macroflora that would address this issue are limited (Erdei et al., 2019). The elements identified to date appear to represent widespread clades in temperate-zone Paleocene floras, such as Platanaceae, Malvaceae, Lauraceae, and rare cycads (Ellis et al., 2003; Erdei et al., 2019). The palynoflora from Castle Rock contains standard early Paleocene taxa for the region, and no distinctive tropical rainforest elements have been reported (Nichols and Fleming, 2002).

## Neotropics

The wet tropical forests of South America, Africa, and Southeast Asia hold most of the Earth's biodiversity (Slik et al., 2015; Pillay et al., 2022); however, the effects of the KPgE in the tropics have been largely unknown. The situation changed with investigations of three outstanding macrofloras in Colombia, from the late Maastrichtian (ca. 68 Ma) Guaduas Formation and the middle-late Paleocene (ca. 60–58 Ma) Cerrejón and Bogotá formations (e.g., Doria et al., 2008; Wing et al., 2009; Correa et al., 2010; Carvalho et al., 2011; Martínez et al., 2015; Herrera et al., 2019). The Paleocene floras represent the world's oldest true Neotropical rainforests, based on family-level composition, leaf traits, and isotopic indicators for closed-canopy environments (Wing et al., 2009; Graham et al., 2019). The large fossil legumes and palm fruits are also typical of tropical forests (Gómez-Navarro et al., 2009; Herrera et al., 2019). The three fossil sites are not as close to the KPgE in age as the others discussed here but far closer than any other tropical localities. Carvalho et al. (2021) recently merged the macrofloral data with a finely resolved palynological dataset from Colombia covering the 72–58 Ma interval, integrating 2,053 Maastrichtian macrofossils, 4,898 Paleocene macrofossils, and 53,029 pollen occurrences of 1,048 taxa from 39 outcrop and well sections. The age estimates rely on well-established biostratigraphic correlations, and preservation

of the K–Pg event horizon in one of the well sections provides a definitive constraint for pre- and post-extinction palynofloras (de la Parra et al., 2022). Notably, the Colombian locations are much closer to Chicxulub (ca. 2000 vs. 3,000 km) than the areas discussed in the western USA and would have frozen during the impact winter according to climate models (Figure 1).

The Carvalho et al. (2021) analysis produced several significant outcomes, most prominently the direct linkage of the KPgE with the emergence of the first tropical rainforests. Pollen data indicated an extinction peak of ca. 45% at 66 Ma, one of the highest recorded globally, significant increases in angiosperm abundance at the expense of gymnosperms, a clear cluster separation of Cretaceous and Paleocene composition, and a ca. 6 Myr recovery of diversity. We note that these palynological results are based on composite stratigraphy, and therefore the extinction estimates are not fully comparable with single, continuous sections. The floristic shift is also striking in the macrofloras. The Guaduas flora contains a generalized mixture of plant families, including Rhamnaceae, Dilleniaceae, and Zingiberales. However, the Paleocene floras have plant-family composition and relative abundance impressively similar to the modern day, dominated by legumes and palms with Melastomataceae, Menispermaceae, Euphorbiaceae, and Malvaceae among the supporting taxa (Figure 3A). The findings from Colombia provide new, detailed evidence that the rise of angiosperms to true ecological dominance, shaping terrestrial biodiversity in the last phase of the Angiosperm Terrestrial Revolution (see Benton et al., 2022), was a direct legacy of the KPgE and the opportunities it created. The results mesh well with a state-of-the-art molecular clock study that used 238 well-constrained calibration fossils (Ramírez-Barahona et al., 2020). Those authors found that angiosperms achieved much of their phylogenetic diversity as stem lineages in the Cretaceous, and crown-group families mostly evolved after the KPgE (Figure 3B).

## Patagonia

Until recently, detailed knowledge of the K–Pg plant extinction in the Southern Hemisphere was mainly limited to palynological work in New Zealand, which showed a fern-spore spike associated with an event horizon and iridium anomaly, ecological disruption, and minimal overall extinction (Vajda et al., 2001; Vajda and Raine, 2003). This situation changed with a series of investigations on Maastrichtian and Danian coastal lowland macrofloras from Chubut, Patagonian Argentina (paleolatitude ca. 50°S); these assemblages were intensively collected and well dated using combinations of radiometric, paleomagnetic, and biostratigraphic constraints (Barreda et al., 2012; Scasso et al., 2012; Clyde et al., 2014; Vellekoop et al., 2017; Clyde et al., 2021). More than 5,000 specimens were collected from the latest Maastrichtian portion of the Lefipán Formation (ca. 66.5–66 Ma) and the Danian Salamanca (ca. 65–64 Ma) and Las Flores (ca. 62 Ma) formations, as well as smaller but systematically informative samples from the Maastrichtian-Danian La Colonia Formation (e.g., Iglesias et al., 2007, 2021; Cúneo et al., 2014). The K–Pg event bed is not preserved in these strata, but the stratigraphic interval that brackets the event or its hiatus is constrained to only a few meters of section in the Lefipán and La Colonia study areas (Barreda et al., 2012; Clyde et al., 2021).

The Patagonian floras have been well-studied systematically. The Lefipán and La Colonia assemblages include Araceae, lotuses, diverse aquatic ferns, and several conifer families (Cúneo et al., 2014; Wilf et al., 2017; Andruchow-Colombo et al., 2018, 2022).

The Salamanca Formation and correlative strata uniquely preserve diverse fossil reproductive structures from the earliest Cenozoic of the Southern Hemisphere (Figure 4), providing extraordinary direct evidence of likely KPgE survivor taxa. Examples include multiple organs of *Agathis*, cocosoid palm fruits (Attaleinae), Rhamnaceae flowers (probable ziziphoid clade), two genera of Cunoniaceae flowers (Schizomerieae and an unknown clade of the family crown-group), and a Menispermaceae (*Stephania*) endocarp (Futey et al., 2012; Jud et al., 2017; Escapa et al., 2018; Jud et al., 2018a,b; Jud and Gandolfo, 2021). Fossil leaves from the Salamanca include species of Podocarpaceae (*Dacrycarpus* and an extinct scale-leaved genus), Akaniaceae, Anacardiaceae, Fabaceae, Lauraceae, Nothofagaceae, and Rosaceae (Quiroga et al., 2016; Andruchow-Colombo et al., 2019; Iglesias et al., 2021). Most of these taxa continued to be prominent elements of later Patagonian fossil assemblages and living Gondwana-derived rainforest floras.

Palynological analysis of the Lefipán, which includes early Danian pollen, provided the only detailed view of K–Pg plant turnover in Patagonia in one continuous section (Barreda et al., 2012). The results included a transient disruption followed by the recovery of nearly all Cretaceous taxa, broadly similar to prior findings from New Zealand (Vajda et al., 2001), and an intriguing spike in the earliest Danian, not of ferns but of the extinct conifer family Cheirolepidiaceae (*Classopollis* pollen). The palynology from all the Patagonian formations shows very similar composition (e.g., Barreda et al., 2012; Clyde et al., 2014; Stiles et al., 2020; Clyde et al., 2021), indicating minimal variation in higher plant taxa (as typically represented by palynomorphs; e.g., Nichols and Johnson, 2008) across the K–Pg or among basins.

An early study of fossil leaves indicated that the Danian Salamanca leaf floras were more diverse than the most comparable Paleocene floras from the western USA (Iglesias et al., 2007). Later, Stiles et al. (2020) produced the first quantitative analysis of macrofloral turnover in the Southern Hemisphere, based on the Lefipán and Salamanca dicot-leaf assemblages ($n > 3,500$ leaves). The authors found a sharp K–Pg drop in rarefied species diversity consistent with an extinction event (Figure 5A). Notably, both the Lefipán and Salamanca samples were more speciose than comparable North American samples (Figure 5A). Only five 'survivor' species were found, suggesting a > 90% face-value extinction far exceeding the North Dakota baseline of 57%. This comparison and the overall decrease in diversity (Figure 5A) strongly support significant species losses but illustrate the importance of age control when comparing extinction estimates. Given its larger age uncertainty, the Lefipán sample could correspond to any of several time slices of the Hell Creek Formation that, if taken alone for strict comparability, would also yield 90% or comparable species extinction (Figure 2B). Morphospace analysis of all leaf species unexpectedly showed that the Danian floras retained and expanded on Cretaceous morphologies, particularly by exhibiting new lobed and toothed leaf forms (Figure 5B). Because the Salamanca floras appear to primarily contain paleo-endemic taxa (Iglesias et al., 2021), this result supported in situ adaptation and evolution in response to an overall cooling trend (Stiles et al., 2020).

## Discussion

The paleobotanical data reviewed here have greatly expanded the understanding of geographic heterogeneity in the K–Pg plant extinction, while illustrating how variations in temporal, spatial, and taxonomic resolution, among other factors, limit comparisons and

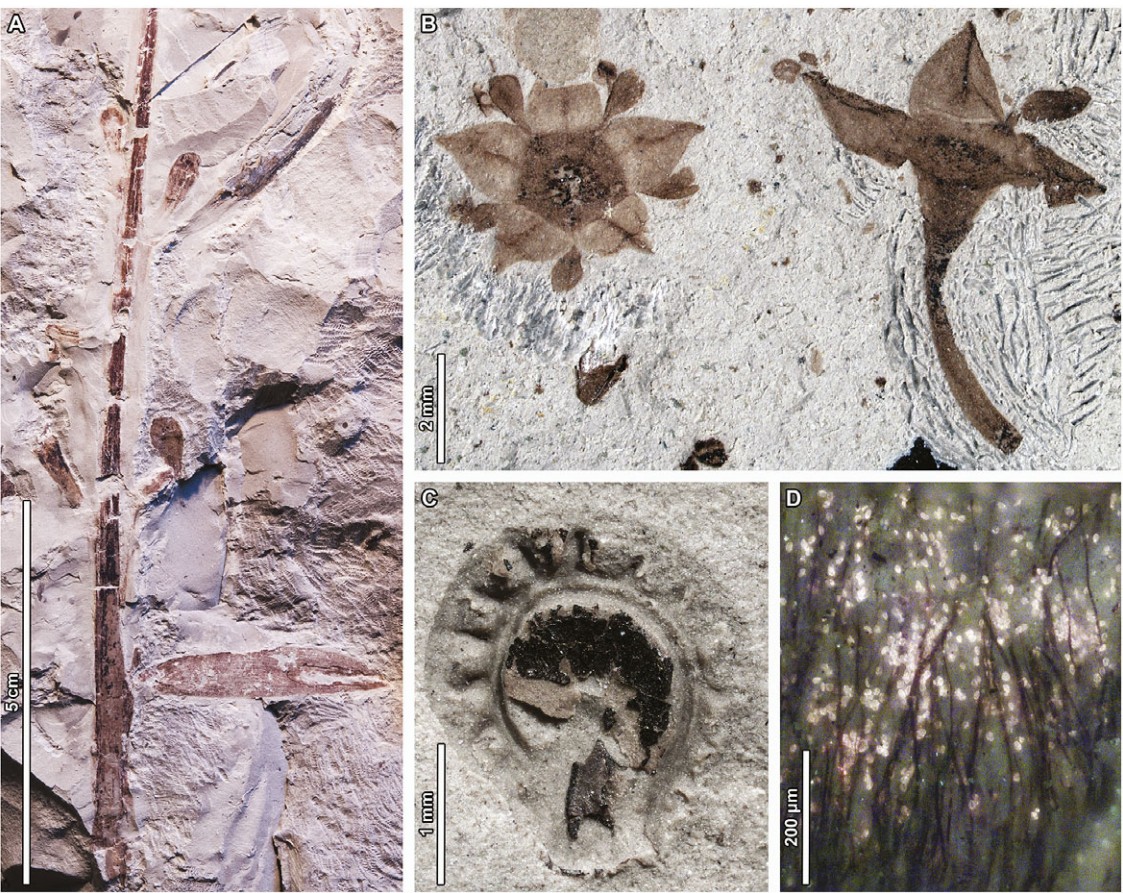

**Figure 4.** Early Danian reproductive fossils from the Salamanca Formation, Chubut, Argentina. (A) *Agathis immortalis* (Araucariaceae) branch with leaves and axillary pollen cones (see Escapa et al., 2018). (B) *Notiantha grandensis* (Rhamnaceae) flowers side-by-side in transverse and longitudinal views (see Jud et al., 2017). (C) *Stephania psittaca* (Menispermaceae) endocarp (see Jud et al., 2018b). *Lacinipetalum spectabilum* (Cunoniaceae) flower, detail of ovary surface covered in trichomes and entrapped pollen grains, glowing under epifluorescence (see Jud et al., 2018a). All the specimens are housed at Museo Paleontológico Egidio Feruglio, Trelew, Chubut. Credits: A, C, D: P. Wilf; B, N. Jud, used with permission.

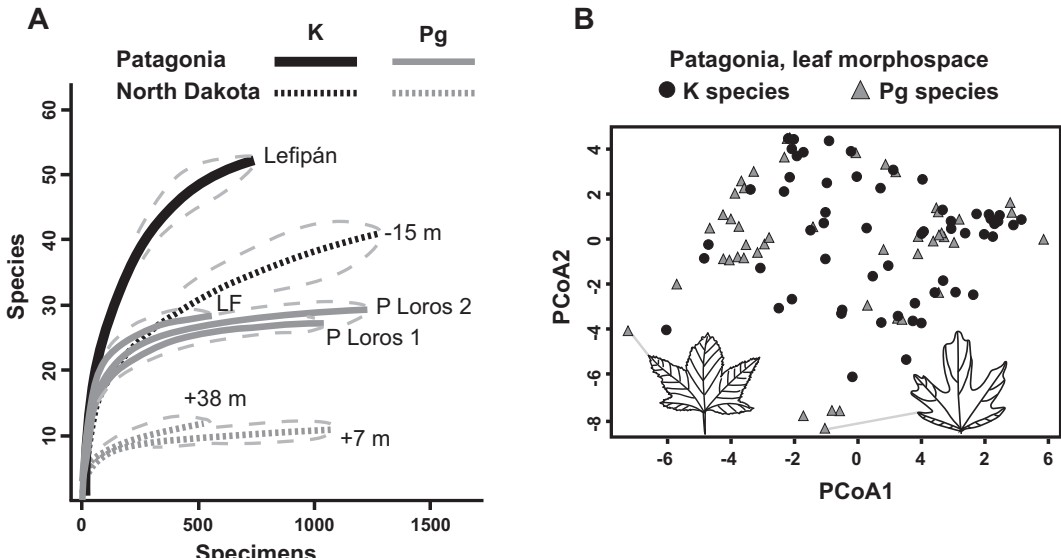

**Figure 5.** Changes in K/Pg leaf diversity and leaf morphospace in Patagonia, modified from Stiles et al. (2020). (A) Rarefied richness (with 95% confidence intervals) for leaf assemblages from the latest Maastrichtian Lefipán, early Danian Salamanca (P Loros 1 and P Loros 2), and late Danian Las Flores formations from Patagonia, compared with three sampling levels from the Williston Basin in North Dakota denoted by their stratigraphic positions relative to the K/Pg event bed (also see Figure 2A). Note the sharp drop in richness in both areas and the higher richness in both time periods for the Patagonian floras. (B) Morphospace plots for Maastrichtian and Danian leaf species in Patagonia, using principal coordinates analysis (PCoA). The Danian leaves have higher morphological diversity, despite lower species richness. Icons and gray lines indicate lobed Danian outlier species and their plot locations.

insights. Despite the advances from the Americas, there are very few well-dated, well-preserved macrofloras of appropriate age elsewhere (Nichols and Johnson, 2008). For example, the most suitable assemblage in Europe is probably the ca. 60–61 Ma Menat Fossil Pit in Puy-de-Dôme, France, which preserves notably diverse plants and insect-feeding damage (Wappler et al., 2009). Much remains to be learned to better trace the evolutionary and biogeographic legacies of the KPgE in the living world flora, and well-constrained paleobotanical data from new areas remain fundamental.

Maritime and latitudinal buffering of impact-winter temperatures may be top-level variables for KPgE severity (Figure 1). Most plant species are frost-intolerant, and the Earth's surface was largely frost-free at the time (Scotese, 2021), suggesting that the terminal Cretaceous vegetation was highly vulnerable to the impact winter. The potential freezing of the tropics is particularly critical for understanding the dramatic turnovers observed in Colombia. The faster recoveries in the southern vs. northern Rockies directly support buffering, whereas the minimal palynological extinctions in Patagonia and New Zealand could reflect both buffering and distance from the crater. The contrasting palynological and macrofloral data from the same strata in Patagonia support the idea (see Introduction) that the K–Pg plant extinction primarily occurred at lower taxonomic levels. Conversely, Colombia's ca. 45% pollen extinction could indicate both drastic species losses and significant elimination of higher taxa.

The end-Cretaceous plant extinction has been questioned because of the apparent lack of significant global losses at the family or major-clade level (Cascales-Miñana and Cleal, 2014; Sauquet and Magallón, 2018; Thompson and Ramírez-Barahona, 2023). However, we hold that the K–Pg event included a massive extinction of plants and more. Estimates of extinction rates based on phylogenies of living taxa are highly uncertain (Louca and Pennell, 2020), particularly towards deeper nodes (O'Meara and Beaulieu, 2021). Attempts to quantify the KPgE should rely on the fossil record. Even though fossil data are probably insufficient to address this question globally and above the species level, every regional macrofossil study with large sample sizes and well-constrained stratigraphy shows significant species losses (usually >50%; Figures 2A, 5A). Species losses are also extinctions, and species conservation is the goal of most modern conservation efforts. Given the wide distribution and high species diversity of angiosperms by the end of the Cretaceous, significant family-level extinctions were probably statistically unlikely. This situation parallels the even more diverse insects, which also show no global extinctions at the family level in large temporal bins bracketing the KPgE (Labandeira and Sepkoski, 1993). However, the disappearances of specialized insect-damage morphotypes on many of the same floras discussed here suggest widespread herbivore losses (Labandeira et al., 2002; Donovan et al., 2014, 2017). The K–Pg event was more than a mass extinction for land plants because it instantaneously transformed terrestrial ecosystems, initiating developments of paramount importance for understanding and conserving today's biotas. These included the radiations and biogeographic movements of the survivor lineages from the KPgE to the present, novel plant–animal interactions, extensive crown-group angiosperm radiations, and the rise of Neotropical and Gondwanan rainforests.

## The future

The KPgE is probably the most relevant deep-time analog for the projected modern-day extinctions, which are also occurring nearly instantaneously in geologic time. Although the worst outcomes are still avoidable (Dinerstein et al., 2020), the coming centuries are projected to bring devastating species and ecosystem losses (Barnosky et al., 2011; Armstrong McKay et al., 2022); these will surely qualify as a mass extinction, even if few plant families disappear. The predictions, although challenging to compare with fossil data, are nevertheless on a scale similar to that of the regional paleobotanical studies reviewed here. For example, a meta-analysis of conservation databases found that extinction currently threatens 39.4% of plant species (Nic Lughadha et al., 2020). If the projected extinctions become a reality, new evolutionary radiations and ecosystems will eventually transform the biosphere as in the geologic past, but most likely far too late to benefit humans.

**Open peer review.** To view the open peer review materials for this article, please visit http://doi.org/10.1017/ext.2023.13.

**Data availability statement.** All data discussed are available in the cited literature.

**Acknowledgements.** We thank Julia Brugger, Santiago Ramírez-Barahona, and Nathan Jud for kindly contributing graphics; the editors and reviewers, including John Alroy, Haijun Song, Vivi Vajda, and one anonymous colleague, for encouragement and helpful comments; and Gabriella Rossetto-Harris, Hervé Sauquet, and Michael Benton for discussions.

**Author contribution.** Conceptual development: P.W., M.R.C., E.S; Writing—original draft: P.W.; Writing—review and editing: P.W., M.R.C., E.S.

**Financial support.** Recent research reviewed here that involved the authors was supported by the National Science Foundation (P.W., Grant Numbers DEB-1556666 and EAR-1925755; M.R.C., Grant Number EAR-1829299), the Smithsonian Tropical Research Institute (M.R.C.), the Geological Society of America (E.S. and M.R.C.), and the Paleontological Society (E.S.).

**Competing interest.** The authors declare none.

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
