## [Reviewer Report]

*Comments to Author*: The manuscript by Wilf et al. provides an interesting overview of published paleobotantical records across the K-Pg boundary in North and South America .The results highlight heterogeneity among records, potentially reflecting a heterogenous response the the K-Pg boundary perturbations.

Overall, the manuscript is well-written, albeit with a somewhat unconventional structure. An example is the unexpected side-discussion on geological records versus molecular clock data (lines 180-192) that is placed a bit awkwardly.

Perhaps the main take-home message is that the paleobotantical records across the K-Pg boundary are actually still rather centered around the Americas, making it difficult to truly assess a global record across the K-Pg boundary. This point could be stressed a bit more in the conclusions.

A few other comments and suggestions:

Lines 75-76:

While the KPME was in all likelihood indeed more rapid than any other geological mass extinction, I would be cautious to use the term ‘severity’ in this sentence, as other mass extinctions (e.g. PT boundary) appear to have resulted in higher overall extinction rates.

Lines 105-106:

I would refrain from using such terminology as “any KPME for plants” and “the plant KPME”, since a mass extinction event is per definition an extinction event across many different biological groups. Hence, one should not say “a mass extinction among plants”, but rather “an extinction event among plants”, of, if you will, “a massive extinction among plants…across the KPg boundary”

Please be careful with this terminology throughout the manuscript.

Line 143:

“70,000 years” gives the suggestion of a level of accuracy that is not really there. Even when adding the “ca.” to this number, it is better to use something like “ca. 70 kyrs”. The alternative option would be to add quantitative uncertainties on this number, but I don’t believe those are available either.

Lines 332-334:

There are a few other interesting sites in Europe, for example the mid-Seelandinian floras of the Gelinden Marls in Belgium, already described in the 19th century by De Saporta et al. and others, where there recently also have been studies on the plant-insect interactions (Zambon et al, 2023).

Lines 374-375: see earlier comments on the terminology around mass extinctions. Please refrain from using terminology such as “a mass extinction for…” followed by a specific biological group.

All in all, these minor points leads me to suggest this manuscript to be accepted for publication with minor revisions.

---

## [Reviewer Report]

*Comments to Author*: This is an interesting and useful overview of the vegetation change and mass extinction related to the end-Cretaceous event 66 million years ago. The text is very well written, easy to follow and keeps the reader interested. It has the North American record in the focus but provides a good global overview supported by figures and diagrams, also including climate effects and the future aspect on the 6th mass extinction. I suggest publication with minor revision.

1. I find the usage of the abbreviation KPME completely unacceptable. P stands for the Permian and should definitely not be used for the Paleogene. If the authors find the need to abbreviate the Cretaceous-Paleogene mass extinction then I suggest K-PgME.

In some places, the authors have used KTME, mainly in the Fig. captions. That is less bad but probably not correct either – please search through the manuscript for these inconsistencies.

2. I lack a short paragraph on holdover taxa in refugia in general. There is data from e.g. Tasmania with plant groups surviving the K-PgME until the Eocene.

3. In the section concerning the impact winter, carbon particles are mentioned as nucleus for aerosols. Importantly also nano-particles of Fe (material dispersed from the asteroid) has shown to play a major role for the following darkness and cooling. https://doi.org/10.1016/j.gr.2014.05.009.

I look forward to see this interesting paper published.

Sincerely

Vivi Vajda

---

## [Editor Report]

*Comments to Author*: Dear Dr. Wilf,

Thank you for submitting your manuscript (ID: EXT-22-0048) to Cambridge Prisms: Extinction. I have now received reports from two reviewers. Collectively, both Reviewers think this paper is well written and significant and should be published. However, as you will see from the reports, the reviewers raised some minor issues that need to be revised or reconsidered, e.g., the abbreviation KPME. Reviewer 1 suggested to raise aspects that need further interpretation, such as the fact that the paleobotanical record across the K-PG boundary is still actually more centered on the Americas, making it difficult to truly assess the global record across the K-Pg boundary. Considering these comments, my decision is a minor revision. 

Please make sure all points raised by the reviewers are addressed and provide a point-by-point response to these comments along with your revision.

Thank you for your understanding and I look forward to receiving the revised manuscript.

Sincerely,

Haijun

Prof. Haijun Song

Handling Editor, Cambridge Prisms: Extinction

Email: haijunsong@cug.edu.cn

---

## [Reviewer Report]

*Comments to Author*: Thank you for addressing all the issues. I look forward to see the paper in print.

Regards

Vivi Vajda

---

## [Editor Report]

*Comments to Author*: Dear Dr. Wilf,

Thank you again for submitting your manuscript (ID: EXT-22-0048.R1) to Cambridge Prisms: Extinction. Your revisions adequately address the concerns raised in the review and I am pleased to confirm that your paper “The end-Cretaceous plant extinction: heterogeneity, ecosystem transformation, and insights for the future” has been accepted for publication in Cambridge Prisms: Extinction.

Yours sincerely,

Haijun

Sincerely,

Haijun

Prof. Haijun Song

Handling Editor, Cambridge Prisms: Extinction

Email: haijunsong@cug.edu.cn